# Multivariate Optimization and Validation of a Modified QuEChERS Method for Determination of PAHs and PCBs in Grilled Meat by GC-MS

**DOI:** 10.3390/foods13010143

**Published:** 2023-12-31

**Authors:** Mahsa Ostadgholami, Mohsen Zeeb, Maryam Amirahmadi, Bahram Daraei

**Affiliations:** 1Department of Applied Chemistry, Faculty of Science, South Tehran Branch, Islamic Azad University, Tehran 1777613651, Iran; mahsaostadgholami@yahoo.com (M.O.); m_zeeb@azad.ac.ir (M.Z.); 2Food and Drug Reference Control Laboratory (FDRCL), Iran Food and Drug Administration (IFDA), Ministry of Health and Medical Education, Tehran 1113615911, Iran; 3Food and Drug Laboratory Research Center (FDLRC), Iran Food and Drug Administration (IFDA), Ministry of Health and Medical Education, Tehran 1113615911, Iran; 4Department of Toxicology and Pharmacology, School of Pharmacy, Shahid Beheshti University of Medical Sciences, Tehran 1996835113, Iran

**Keywords:** central composite design, contaminants analysis, gas chromatography–mass spectrometry, grilled meat, PAHs, PCBs

## Abstract

Polycyclic aromatic hydrocarbons (PAHs) and polychlorinated biphenyls (PCBs) are recognized as carcinogens and mutagenic food contaminants that threaten public health. As for food safety aspects, control of these contaminants in processed and fatty food is necessary. In this study, eleven factors were screened by the Plackett–Burman design, and four variables were chosen to optimize with the central composite design (CCD) for the improvement of extraction and cleanup procedures of these food contaminants. The optimized variables include 5 g of sample, 2 mL mixture of 2/2/1 ethyl acetate/acetone/isooctane, 1.6 g of ammonium formate, 0.9 g of sodium chloride, and 0.25 g of sorbent Z-Sep+. A 5 min cleanup vortex time with the spike calibration curve strategy, analyzed by gas chromatography–mass spectrometry (GC-MS), led to the validated limits of quantification (LOQs) for 16 PAHs and 36 PCBs of 0.5–2 and 0.5–1 ng/g, respectively, and recoveries of 72–120%, with an average relative standard deviation (%RSD) of 17, for PAHs, and 80–120%, with an %RSD of 3, for PCBs. The method introduces excellent accuracy, precision, and efficiency, and minimizes matrix effects, and ensures a control procedure, adopted with international standards, for food authorities to determine the contaminants of interest in processed meat, and consequently, prevent food-borne disease to improve public health indices.

## 1. Introduction

Persistent organic pollutants (POPs), as compounds of anthropogenic and natural (bacterial and plant) sources, have high lipophilicity, a determinant of bioaccumulation in living organisms’ fatty tissues. Two groups of these compounds are polycyclic aromatic hydrocarbons (PAHs) and polychlorinated biphenyls (PCBs), and their persistence in the environment is a critical issue due to their toxic effects [1,2,3,4,5,6,7]. Therefore, these compounds are included in the priority pollutant list.

Polycyclic aromatic hydrocarbons (PAHs) are categorized as lipophilic compounds consisting of two or more fused aromatic rings. PAHs containing four rings, such as benzo(a)anthracene and chrysene, are weak carcinogenic compounds. In addition, heavyweight PAHs with more than five rings, such as dibenz(a,h)anthracene, benzo(a)pyrene, indeno(1,2,3-cd)pyrene, benzo(b,k)fluoranthene, and benzo(g,h,i)perylene, might have genotoxic and carcinogenic properties harmful to humans, and therefore, are considered as potent contaminants, and are potentially a serious public health concern.

PAHs are ubiquitous environmental contaminants, and can be produced during food processing. Also, they may originate in environments with natural and anthropogenic sources, industrial food processing (such as heating, drying, and smoking), packaging, and certain cooking practices (grilling, roasting, and frying). Food and diet are the main sources of PAH exposure for non-smokers and non-occupationally exposed adults [8,9,10,11].

PCBs are long-lasting pollutants that result from the incomplete combustion of organic and synthetic substances. They are found in most industrial and consumer products. The production of PCBs was banned in the United States in 1977. Because of their persistence and bioaccumulation, PCBs are categorized as POPs. The major consumer health concern is the presence of PCBs in industrial products such as plasticizers in paints, plastics, and rubber materials. PCBs accumulate in fatty tissue, so are typically found in foods of animal origin. Exposure to PCBs is commonly caused by the intake of fish; however, other foods, such as beef, dairy, and chicken products, with lower PCB levels are more frequently consumed, so are also considered PCB sources. The adverse effects of PCBs in humans are cancer, immune system disorders, reproductive organ dysfunction, and disorders of the neurological and other biological systems. PCBs are recognized as pollutant markers because of their presence in biotic and abiotic conditions [12,13,14,15].

Previous studies have reported the high levels of carcinogenic compounds produced in grilled foods like meat, meat products, fish, or others compared to foods prepared via other cooking methods [16].

Among the types of foods, the consumption of grilled, barbecued, and smoked meat is the main means of exposing the body to PAHs. The presence of PAHs in meat can result from the thermal decomposition of organic matter in the meat, as well as from smoke from the incomplete combustion of fat in meat drippings that drip over the fire during grilling using charcoal [17,18]

These foods are attractive both in restaurants and at home; nonetheless, they pose a great health risk to consumers [16].

The European Commission first established maximum levels (MLs) for benzo[a]pyrene (BaP) in several food types, including smoked meat and fish, in 2005 (Commission Regulation (EC), 2005). JECFA re-evaluated PAHs in 2005 (Joint FAO/WHO Expert Committee on Food Additives, 2005), and concluded that 13 PAHs are clearly genotoxic and carcinogenic, and 2 other PAHs may contribute to the formation of lung tumors. This gave a group of 15 PAH compounds (PAH15, and if the substituted PAH, 5-methyl chrysene, is included, PAH15 + 1). In 2008, EFSA published an opinion on PAHs (EFSA, 2008), and concluded that BaP alone was not a suitable general marker for PAHs in food, but identified a group of four PAHs (PAH4) and a group of eight PAHs (PAH8) as better indicators based on data relating to occurrence and toxicity. Measuring PAH8 offered little additional benefit compared with PAH4. Based on the EFSA’s opinion, in 2011, the European Commission extended the scope of the regulation to include other types of food, and to add limits for PAH4 (Commission Regulation (EU), 2011) [19,20].

For food quality evaluation, the European Commission, for food and animal products, sets maximum limits (MLs) for PAHs and PCBs [21,22]. Because of the low level of established MLs by legislative agencies, it is necessary to expand practical, accurate, and sensitive analytical procedures to determine these compounds in different matrices. Multicontaminant analytical methods are used in monitoring and controlling food samples, especially for the determination of PAHs and PCBs, including the following techniques: gas chromatography–mass spectrometry (GC-MS) [23,24], GC-flame ionization detection (GC-FID) [25], GC-electron capture detection (GC-ECD) [1,26], gas chromatography–tandem mass spectrometry (GC-MS/MS) [1,27,28], liquid chromatography–tandem mass spectrometry (LC-MS/MS) [29,30], gas chromatography–high-resolution mass spectrometry [31,32], and supercritical fluid chromatography/atmospheric pressure chemical ionization–mass spectrometry (SFC/APCI-MS) [33]. GC-MS can identify, separate, and quantify trace components in many complex compounds. Also, GC-MS is an efficient analysis technique for multiple contaminants with thermal stability, less polar characteristics, and higher volatility [23].

The sample pretreatment process includes several cleanup steps to eliminate the unwanted co-extraction of the sample, which can be time-consuming and labor-extensive. Several sample pretreatment methods are used to determine PAHs and PCBs in food, such as micro-matrix solid-phase dispersion (micro-MSPD) [1], solid-phase micro-extraction (SPME) [34], gel permeation chromatography (GPC) [31], supercritical fluid extraction (SFE) [35], and QuEChERS (quick, easy, cheap, effective, rugged, and safe).

The QuEChERS method aims to simplify and streamline the extraction and purification procedures and minimize cost, and it includes miniaturization and automation to make the analysis easier, while the conventional extraction methods require multiple steps and a considerable amount of solvent and time [36].

Nowadays, the main advantages of QuEChERS, such as the effective elimination of the matrix effect as well as the capacity to obtain a high recovery of target analytes, are extensively known in diverse fields, such as food, environmental, and biological analysis [37].

Moreover, the great versatility of the procedure has allowed its application in other types of matrices and analytes with excellent results [38].

Various modified QuEChERS methods were applied to determine contaminants, such as PAHs and PCBs, in different matrices. Some of these matrices include animal-originated foods, like fish and seafood [39], ham [24], poultry meat [40], eggs [41], and milk [42], as well as rice [43] and vegetable oil [44]. C18 and primary-secondary amine (PSA) are repeatedly applied as common dSPE sorbents to increase the cleanup performance in specimens with ≥2% fat; new dSPE sorbents are also available, such as hyper-cross-linked polystyrene, metal–organic frameworks [45], diatomaceous earth [46], graphene, molecularly imprinted polymers [47], and multiwalled carbon nanotubes [48]. These have also been applied to increase the effectiveness of sample preparation for various contaminants and matrices. Among the mentioned methods, Z-Sep+ and efficient matrix removal-lipid (EMR-Lipid) are two new commercial sorbents used for the cleanup of fat samples. Z-Sep+ is made of two sorbents, octadecyl (C18) and silica coated with zirconium dioxide (ratio: 2/5) [49]. Using Z-Sep+ is an approach to combine the chemical properties of zirconia with the high surface area of C18. Zirconium is a Lewis acid that interacts strongly with Lewis bases such as phospholipids [50].

The EMR-Lipid structure is a proprietary secret, and its mechanism has hydrophobic interactions and size exclusion. The producer claims that EMR-Lipid can selectively remove lipids from QuEChERS extracts of fatty foods in all cases [2]. During the extraction, a high surface with elevated absorption capacity and low specific weight is applied as a dispersant.

In this study, our purpose was to improve validation parameters by selecting extraction sorbents and solvents and optimizing the content of salts, solvents, sorbents, and time for extraction and cleanup in the QuEChERS method. The used meat matrix was sheep heart, one of the most nutritive, popular, and delicious grilled foods in Iran. 

In dSPE methods, optimizing the condition to achieve maximum efficiency is essential. Statistical optimization techniques, such as response surface methodology (RSM), can anticipate and indicate optimal conditions of the favorable process using validated models [51,52]. Such a method is more efficient and valuable than a univariate optimization method, called one-variable-at-a-time (OVAT), with fewer experimental runs [53].

The major disadvantage of the OVAT approach is that it cannot investigate the interaction among factors. RSM-based methods can be considered the best solution to overcome this problem and obtain optimal experimental conditions.

In the first step, independent variables, including solvent type and its volume, vortex time, and the amounts of sample, salts (ammonium formate and sodium chloride) [54], and sorbents, (PSA, Z-Sep+, and EMR-Lipid), were investigated using the Plackett–Burman two-level fractional factorial design. Then, the central composite design (CCD) was used to optimize the QuEChERS important variables for simultaneous GC-MS determination of 16 PAHs and 36 PCBs in grilled sheep heart meat.

## 2. Material and Methods

### 2.1. Chemicals—Reagents

PAHs, PCBs, and their isotope-labeled internal standard were prepared by Dr. Ehrenstorfer GmbH (Augsburg, Germany). The purity of standards ranged from 95 to 99.8%. Solvents (acetonitrile, acetone, and isooctane) of HPLC grade, sodium chloride (NaCl), and anhydrous magnesium sulfate (MgSO_4_) of analytical grade were obtained from Merck (Darmstadt, Germany). Also, primary secondary amine (PSA) was obtained from Agilent (Santa Clara, CA, USA), ammonium formate (NH_4_O_2_CH) was supplied by Fluka (Dresden, Germany), Z-Sep+ was supplied by Supelco (Bellefonte, PA, USA), and EMR-Lipid was purchased from Agilent (USA). The intermediate, stock, and working standard mixed solutions of PAHs and PCBs at 100 ng/mL, and internal standard solution at 10 μg/mL, were obtained in acetonitrile and isooctane, respectively, and kept at −18 °C with protection from light.

### 2.2. Apparatus and Instruments

The GC-MS chromatographic analysis of PAHs and PCBs was conducted using an Agilent network GC system 6890N, 5975B Inert MSD (Wilmington, NC, USA) equipped with the 7683B autosampler system Agilent Technologies, (Wilmington, NC, USA). An HP-5MS column (30 m × 0.25 mm id. × 0.25 μm) was used for analysis. Helium was applied as the carrier at 1.0 mL/min. For analyzing PAHs and PCBs in GC-MS, the following GC conditions were used. The oven was set to 75 °C for 3 min, and then the temperature was elevated to 120 °C at 25 °C/min. After that, the oven was heated to 280 °C at 5 °C/min, and remained at this temperature for 21 min. The process was performed at 3 μL injection volume in splitless mode. The electron ionization energy, ion source temperature, and transfer line temperature were 70 eV, 230 °C, and 280 °C, respectively. The total analysis duration was 61.80 min. The screening assessment was conducted based on the selected ion monitoring mode, and a minimum of three characteristic ions were selected for each compound. A centrifuge Eppendorf (Hamburg, Germany) was used at 5816 rcf, and the mixing steps were carried out using a Multi Reax shaker Heidolph, (Schwabach, Germany). Ultra-pure water was obtained from a Millipore (Direct-Q3UV) water purification system (Darmstadt, Germany).

### 2.3. Modified QuEChERS Sample Preparation

The grilled samples were homogenized before the extraction and cleanup process. The extraction steps were performed as follows. Five grams of the specimen was weighed in a 50 mL Falcon tube, and the blank sample was spiked with a combination of PAH and PCB standards at 100 μg/kg. Then, 20 μL at 10 μg/g of isotopically labeled mix 111, 118, and 188 for PCBs and perylene, chrysene, and pyrene for PAHs was added as the internal standard. Next, 2 mL of mixed solvents, including ethyl acetate + acetone + isooctane (2:2:1) (*v*/*v*/*v*), and 500 µL of deionized water was transferred to the tube, and the mixture was shaken strongly for 60 s. Then, 0.9 g of NaCl and 1.6 g of ammonium formate were added to the mixture, which was mixed in a Multi Reax vortex for 15 min, and centrifugation was conducted at 5816 rcf for 10 min at room temperature. Afterward, the supernatant without a fat layer was added to a 15 mL Falcon tube containing 0.250 g of Z-Sep+ and 0.1 g of PSA, followed by shaking for 5 min and centrifuging at 5816 rcf for 10 min at room temperature. After cleanup, the whole upper phase was added to a vial for GC injection. In comparison with the main QuEChERS method, using the solvent mixture, adding ammonium formate, and applying Z-Sep+ are considered modifications of the pretreatment method.

## 3. Results and Discussion

### 3.1. Optimization of the Modified QuEChERS (Sample Preparation) Technique Using Experimental Design

At first, by performing the Plackett–Burman design (PBD), factors with highly significant effects were identified. Table 1 indicates the primary factors and their levels and notations. 

The sum of the 52 peak areas in various positions as the most intense peaks relative to the maximum peak in TIC was selected as a response. Table 2 shows the PBD parameter from 11 variables with 24 experimental runs using Design-Expert software (Design-Expert, Stat-Ease, Minneapolis, MN, USA). 

Sample weight (A), type of solvent (B), solvent volume (C), water volume (D), NH_4_O_2_CH. weight (E), PSA weight (F), type of sorbent (G), amount of sorbent (H), extraction vortex time (J), cleanup vortex time (K), and NaCl weight (L) were considered essential variables. After the screening design, the changed QuEChERS method was optimized by considering the four most significant variables. Therefore, CCD was conducted with four quantitative variables established as 0.5–2.5 g of NH_4_O_2_CH (C), 0.1–0.3 g of sorbent weight (D), 0.1–1.5 g of NaCl weight (E), and 60–600 s for the cleanup vortex time (H). The design matrix of CCD optimization is shown in Table 3. A total of 36 experimental procedures were performed randomly using a quintet at the central point.

### 3.2. Method Validation 

The proposed strategy (modified QuEChERS in combination with GC-MS) was validated based on the validation criteria for quantitative approaches for pesticide assessment in food and feed in the SANCO document 2021/11312 [55]. The linearity, limit of quantification, accuracy, and precision factors were evaluated based on this instruction.

### 3.3. Optimization Process Screening Design, Plackett–Burman Design, and Experimental Design Approach

For achieving efficient specimen pretreatment, and thus increasing recoveries for all analyses, primary investigations were determinative in the choice of extraction factors. We used PBD to determine the significance of 11 variables influencing the changed QuEChERS approach for the 16 PAHs and 36 PCBs. Figure 1 shows the GC chromatogram of the targeted PAHs and PCBs. Experimental results acquired by PBD are illustrated in Table 2 (Response).

According to the ANOVA of the evaluated model, the significant variables were distinguished, which are presented in Table 4. 

Table 5 shows fit statistics parameters for PBD with excellent values. Also, half-normal probability plots were applied to find the most significant interactions and impacts. 

In half-normal probability plots, the negligible impacts are in a straight line, while essential impacts are off the line. The half-normal probability plots for PBD are displayed in Figure 2. Another way to recognize significant main and interaction effects is using the Pareto plot. 

It specifies an absolute value for different effects and determines reference lines at a defined confidence limit (based on the t-value and Bonferroni limit). Any effect that passes these defined lines is considered a significant effect (Figure 3). The t-value, sample weight, sorbent weight, NaCl weight, cleanup vortex time, type of solvent, water volume, and ammonium formate weight were regarded as essential factors.

As the type of organic solvent significantly affects the extraction process, this factor was also studied. It is concluded that the highest response for the analyses was acquired with a mixture of acetone + ethyl acetate + isooctane (2:2:1) (*v*/*v*). The log Kow for PAHs applied in the current research ranged from 3.3 for naphthalene to 6.63 for benzo[ghi]perylene. Acetonitrile presented log Kow = −0.34, whereas amounts for acetone, ethyl acetate, and isooctane were, respectively, −0.24, 0.73, and 4.1. Our finding suggests that the three-component extraction solvent was applied in the extraction method.

Extraction with ethyl acetate isooctane and acetone at the molecular level can result in amended recoveries by letting water-solvable ethyl acetate and acetone recover PAHs trapped in water and making these compounds present for transfer to isooctane. High extraction temperatures can disrupt analyte matrix interactions by decreasing the activation energy needed for analyte desorption processes and decreasing solvent viscosity, simplifying more favorable solvent matrix penetration [56].

It should be pointed out that Z-Sep+ and EMR-Lipid functioned similarly in obtaining chromatograms with no interfering peaks. However, we chose Z-Sep+ because it has less volume than EMR-Lipid in similar weight.

Since the maximum sample weight and the mixture of solvents maximize the defined responses and have the same trend in all responses, and also due to avoiding time consumption and the use of dangerous chemical materials, these factors were not considered for the optimization phase. The reason is that increasing each variable and designing a model for optimization forms extra experimental runs. It means that 5 g of the sample (considering the matrix effect) and a mix of ethyl acetate, acetone, and isooctane have been selected as the optimum conditions.

Also, the water volume was excluded as it had a lower impact than other factors, and its t-value was close to the reference value. Finally, NaCl (A), sorbent weight (B), ammonium formate weight (C), and cleanup vortex time (D) were considered in the optimization step (Table 6).

### 3.4. Central Composite Design

After determining the critical factors, i.e., (NaCl (A), sorbent weight (B), ammonium formate (C), and cleanup vortex time (D), they were optimized by CCD at 5-level three blocks with a total of 36 experiments. Table 3 shows the response of the CCD experiment. According to different statistical parameters of the model, such as error and precision, the quadratic model was selected as the best one. Equation (1) shows the coefficients of the factors and their interactions selected by the model.
Response (recovery) 1.31 = 242.59 + 15.66A + 10.18B − 0.0646C − 14.08D − 10.76AB + 12.67AC − 8.26AD + 2.22BC + 7.16BD + 3.04CD − 15.21A2 − 4.77B2 − 13.66 C2 − 29.83D2(1)

Table 7 shows that the recommended model is highly important, considering its F-value (F model = 139.79 > F Lack of fit = 1.29) and probability value (*p* = 0.0001 < 0.05). Also, the fit derived from the anticipated model is lower than the calculated F value.

Table 8 shows the statistical parameters for CCD, indicating good model fitting in terms of calibration and prediction.

An excellent value of adjusted R^2^ (0.98) indicates that the model is not overfitted. In addition, a good value of predicted R^2^ (0.95) shows that the model has a high capability to predict unknown points. Also, the plot of actual versus predicted R^2^ (Figure 4) shows the quality of the developed model. The points should be close to the fitted line for a proper fit.

As it is clear from the figure, all points are appropriately fitted in the model line, which confirms the quality of the developed model. Therefore, the model is statistically valid and significant. The coefficients related to independent variables based on the evaluated variables are presented in Equation (1). 

According to the equation, NaCl weight (A), sorbent weight (B), cleanup vortex time (D), and interactions between A and C, B and C, B and D, and C and D affect the extraction effectiveness significantly. The coefficients’ negative values belonging to D, AB, AD, A^2^, B^2^, C^2^, and D^2^ indicate their negative impact on the recovery of contaminants from the specimens. The contaminant recovery is reduced with an increase in their values. Also, the terms A, B, BC, BD, CD, and AC positively affect the recovery value. 

The model accuracy was controlled by comparing the anticipated and detected experimental findings, indicating the linear association between the experimental recovery and the anticipated values. 

### 3.5. Response Surface Graphs by CCD and Optimal Values

Figure 5 Show the desired responses concerning the selected factors and interactions.

The graphs show the impacts of two variables in their evaluated ranges, and another variable was fixed to the optimum level. The response surface can better display the trend of the variables and their impact on extraction efficiency. Therefore, the achieved recoveries in the extraction process are influenced by target parameters, and the graphs’ nonlinearity confirms their interactions. The optimization function was used to achieve an optimal situation for the modified QuEChERS process. The desirability function is an excellent tool for simultaneously optimizing multiple responses. This function works according to the logic that a response is acceptable when its attributes are in the desired range. This process combines all responses into a single one, followed by its optimization. The value of the desirability function varies from 0 (undesirable value) to 1 (desirable value). The response is ideal if this value is closer to 1. Figure 6 shows the chosen solution according to the maximum obtained desirability (0.82).

Finally, NaCl weight, sorbent, ammonium formate, and cleanup vortex time were obtained at 0.9 g, 0.25 g, 1.6 g, and 5 min, respectively. Sample amount (A), type of solvent (B), solvent volume (C), water volume (D), type of sorbent (G), and extraction vortex time (J) were chosen, amounting to 5 g, a mix of ethyl acetate + acetone + isooctane (2:2:1), 2 mL, 500 µL, Z-Sep+, and 15 min, respectively. We used the GC-MS analysis method to simultaneously measure the 16 PAHs and 36 PCBs in grilled meat samples. The compounds, quantification ions, and their retention times are shown in Table 9.

### 3.6. Method Validation and Performance

Following the specification of the optimum experimental situation, the suggested method was validated according to the SANCO guidelines regarding range and linearity, LOQ, intermediate precision, and accuracy.

### 3.7. Linearity and Limits of Quantification

The target method linearity was studied using spike calibration curves. Seven blank matrix samples were spiked with mixed standard working solutions at seven levels ranging between 0.5 and 40 μg/kg in triplicate (pretreated), and experimented upon using the modified QuEChERS method explained earlier. The coefficient of determination (R^2^) for the spike calibration curves was achieved at 0.999. The results display appropriate linearity for the contaminant residues in the cited matrices within the considered range. LOQs for the PAHs and PCBs were achieved at 0.5–2 ng/g and 0.5–1 ng/g, respectively, according to the lowest level of the spiked samples. The recoveries and RSDs were accepted with a range between 70 and 120% and below 20%, respectively, for LOQ. Regarding MRLs, established by the European Union for PAHs and PCBs in grilled meat, all the LOQ values in grilled meat materials were less than the MRL values. Hence, the studied method is appropriate for detecting contaminants in grilled meat (Table 9).

### 3.8. Accuracy and Precision

The repeatability intraday and interday (three various days) precision were evaluated through %RSDs of recovery studies from the mean of five replicates of the spiked blank matrix at 0.5, 1, 2, 4, 7, 15, and 30 μg/kg levels in grilled meat. The recoveries were between 72 and 120% and 80 and 120% for the PAHs and PCBs, respectively. Also, the %RSD values were below 17% for the repeatability of the PAHs and 3% for the PCBs. Therefore, the changed QuEChERS method was sufficiently sensitive, accurate, and precise to determine simultaneously the assessed contaminants in the grilled meat samples. The linear range, LOQ values, coefficient of determination, recovery, and RSD are shown in Table 9.

Our proposed methods have some advantages, such as low consumption of the solvent, high recovery, short extraction time, no matrix interference, and good merit figures, compared to the other methods. Also, it should be noted that to eliminate the matrix effect, the calibration curve is drawn based on the spike calibration method, and the range of linearity is from 0.5 to 40 ng/g. Recoveries and RSD data of the analytical method met the criteria as set by European Commission Regulation No. 1881/2006. Also, the LOQ value was investigated at the levels of 0.5–40 ng/g, and after evaluating the accuracy and precision and meeting the criteria, it was determined as the LOQ of the method, and it is not only based on 10 times the S/N ratio. The comparison of this study with other methods was shown in Table 10.

## 4. Conclusions

We attempted to further develop a sensitive and effective method for the simultaneous determination and extraction of trace amounts of a mixture of PAHs and PCBs, as common food contaminants, from grilled sheep heart meat. The modified QuEChERS method was successfully used for high-performance extraction of the analytes of interest before determination by GC-MS in grilled meat. The multivariate optimization technique employing response surface methodology according to CCD was performed to optimize effective factors determining the efficiency of the modified QuEChERS method. The findings demonstrate that the proposed method has many superior features, such as a high enrichment factor, good repeatability, high recovery, short pretreatment time, good precision, and favorable figures of merits, as well as lower LOQs to acquire sensitivity in accordance to the regulation set by international standards. The present method is applicable for food authorities to control and analyze contaminants up to the MLs of the European Commission for food and animal product collections for simultaneous determination of PAHs and non-dioxin-like PCBs in all types of grilled meat matrices, to prevent food-borne diseases and improve public health indices.

## Figures and Tables

**Figure 1 foods-13-00143-f001:**
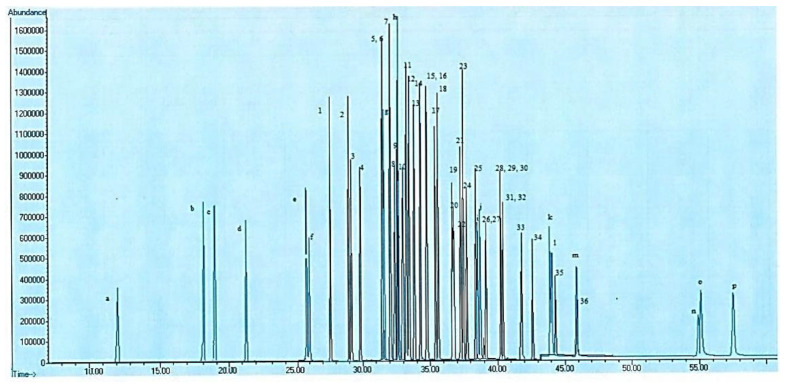
GC chromatograms of PAHs (a–p) of interest and PCBs (1–36) of interest that are referred to in Table in Section 3.5.

**Figure 2 foods-13-00143-f002:**
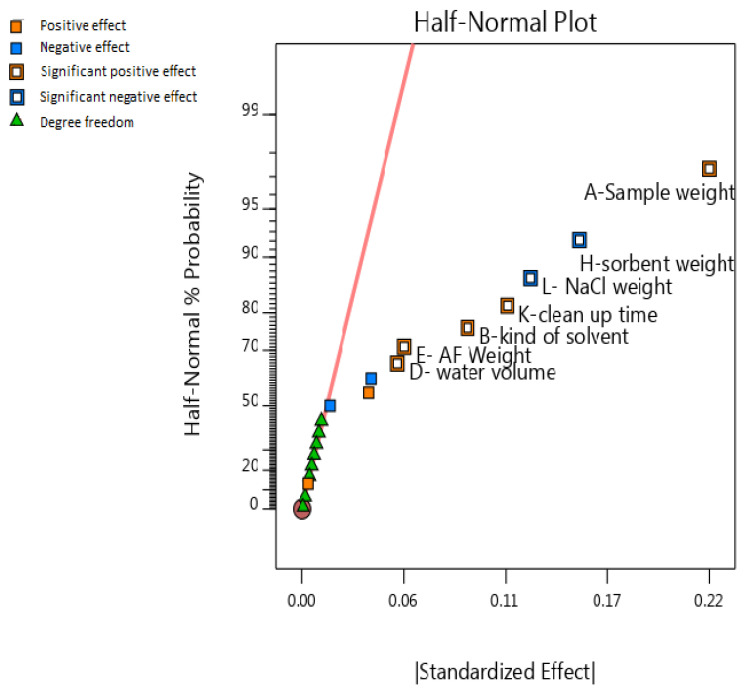
The half-normal plot of factors. Factors that are far from the line are significant in the experiment.

**Figure 3 foods-13-00143-f003:**
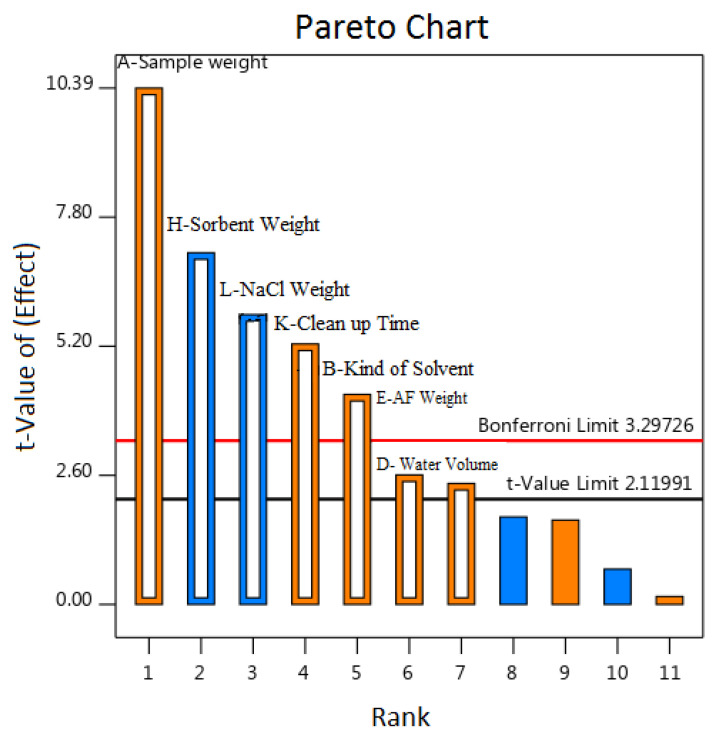
Pareto plot for screening factors in Plackett–Burman design.

**Figure 4 foods-13-00143-f004:**
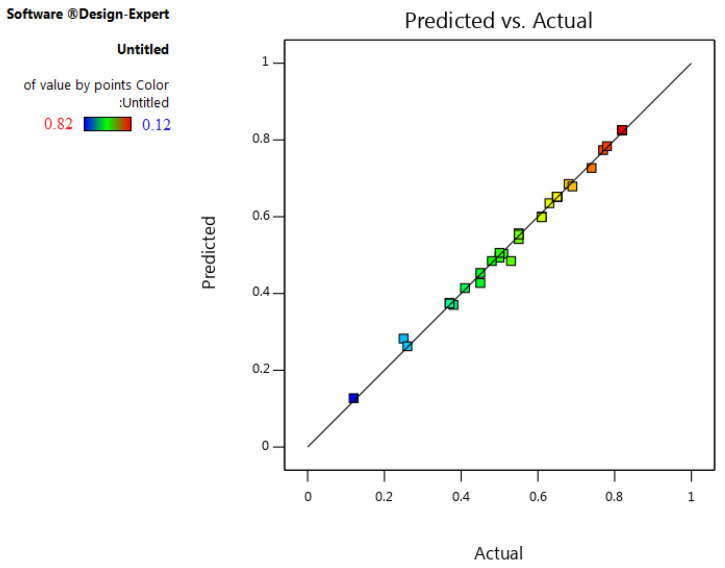
Predicted versus actual R^2^ plot for CCD optimization.

**Figure 5 foods-13-00143-f005:**
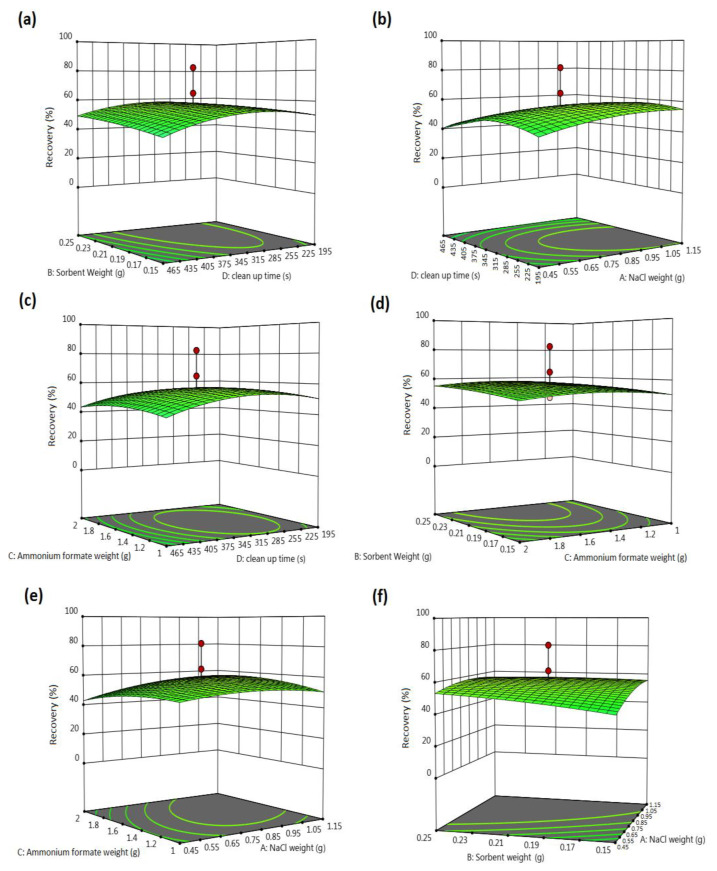
(**a**–**f**) Response surface 3D plots of CCD design.

**Figure 6 foods-13-00143-f006:**
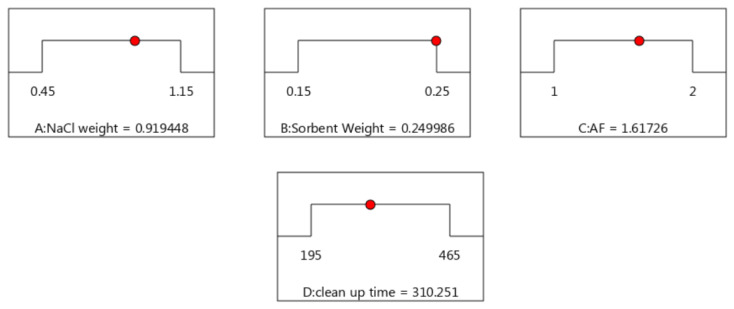
Optimized values of the considered factors in CCD.

**Table 1 foods-13-00143-t001:** Factors, factor notation, type, unit, and their levels in Plackett–Burman design (PBD).

Factor	Notation	Type	Unit	Levels	Factor
				−1 *	+1 **
Sample weight	A	Numeric	g	1	5
Solvent volume	C	Numeric	mL	1	3
Water volume	D	Numeric	mL	100	1000
Ammonium formate weight	E	Numeric	g	0.5	2
PSA weight	F	Numeric	g	0.1	0.3
Amount of sorbent	H	Numeric	g	0.1	0.3
Extraction vortex time	J	Numeric	s	60	1800
Cleanup vortex time	K	Numeric	s	60	900
NaCl weight	L	Numeric	g	0.1	2
				Levels One	Levels Two
Type of solvent	B	Categoric		Acetonitrile	Mix of solvent
Type of sorbent	G	Categoric		Z-Sep+	EMR-Lipid

* −1: Minimum; ** +1: Maximum.

**Table 2 foods-13-00143-t002:** Plackett–Burman parameter with the 11 independent variables investigated.

RUN	A	B	C	D	E	F	G	H	I	G	K	Response
1	3	Acetonitrile	2	550	1.25	0.2	EMR-Lipid	0.2	930	480	1.05	0.1908
2	1	Acetonitrile	1	1000	0.5	0.3	EMR-Lipid	0.1	1800	900	2	0.01788
3	3	Acetonitrile	2	550	1.25	0.2	Z-Sep+	0.2	930	480	1.05	0.2591
4	5	Mix of solvents	1	1000	2	0.3	Z-Sep+	0.1	60	900	0.1	1
5	3	Mix of solvents	2	550	1.25	0.2	EMR-Lipid	0.2	930	480	1.05	0.6712
6	3	Acetonitrile	2	550	1.25	0.2	EMR-Lipid	0.2	930	480	1.05	0.2749
7	1	Acetonitrile	1	100	0.5	0.1	Z-Sep+	0.1	60	60	0.1	0.0324
8	5	Mix of solvents	3	100	0.5	0.1	EMR-Lipid	0.1	1800	900	0.1	0.8902
9	5	Mix of solvents	1	100	0.5	0.3	Z-Sep+	0.3	1800	60	2	0.0089
10	1	Mix of solvents	3	100	2	0.3	EMR-Lipid	0.1	60	60	2	0.0671
11	3	Acetonitrile	2	550	1.25	0.2	Z-Sep+	0.2	930	480	1.05	0.3245
12	3	Mix of solvents	2	550	1.25	0.2	EMR-Lipid	0.2	930	480	1.05	0.7444
13	1	Mix of solvents	1	1000	2	0.1	EMR-Lipid	0.3	1800	60	0.1	0.0847
14	3	Mix of solvents	2	550	1.25	0.2	Z-Sep+	0.2	930	480	1.05	0.7397
15	5	Acetonitrile	3	1000	0.5	0.3	EMR-Lipid	0.3	60	60	0.1	0.2305
16	5	Acetonitrile	1	100	2	0.1	EMR-Lipid	0.3	60	900	2	0.2176
17	1	Acetonitrile	3	100	2	0.3	Z-Sep+	0.3	1800	900	0.1	0.0322
18	5	Acetonitrile	3	1000	2	0.1	Z-Sep+	0.1	1800	60	2	0.4142
19	3	Acetonitrile	2	550	1.25	0.2	EMR-Lipid	0.2	930	480	1.05	0.3290
20	1	Mix of solvents	3	1000	0.5	0.1	Z-Sep+	0.3	60	900	2	0.0541
21	3	Mix of solvents	2	550	1.25	0.2	Z-Sep+	0.2	930	480	1.05	0.7905
22	3	Mix of solvents	2	550	1.25	0.2	Z-Sep+	0.2	930	480	1.05	0.7707
23	3	Mix of solvents	2	550	1.25	0.2	EMR-Lipid	0.2	930	480	1.05	0.7778
24	3	Acetonitrile	2	550	1.25	0.2	Z-Sep+	0.2	930	480	1.05	0.3966

**Table 3 foods-13-00143-t003:** Experimental design matrix for CCD.

Block	Run	NaCl Weight (g)	Sorbent Weight (g)	Ammonium Formate Weight (g)	Cleanup Time (s)	Recovery%
Block 1	1	0.45	0.15	1	465	61
Block 1	2	1.15	0.15	1	195	77
Block 1	3	0.45	0.25	2	465	69
Block 1	4	0.45	0.25	1	195	74
Block 1	5	0.8	0.2	1.5	330	82
Block 1	6	0.8	0.2	1.5	330	82
Block 1	7	1.15	0.25	2	195	78
Block 1	8	0.8	0.2	1.5	330	82
Block 1	9	0.8	0.2	1.5	330	82
Block 1	10	0.45	0.15	2	195	61
Block 1	11	1.15	0.25	1	465	65
Block 1	12	0.8	0.2	1.5	330	82
Block 1	13	1.15	0.15	2	465	68
Block 2	14	0.45	0.25	2	195	50
Block 2	15	0.8	0.2	1.5	330	65
Block 2	16	0.45	0.15	2	465	38
Block 2	17	0.45	0.15	1	195	51
Block 2	18	0.45	0.25	1	465	55
Block 2	19	1.15	0.15	2	195	63
Block 2	20	0.8	0.2	1.5	330	65
Block 2	21	0.8	0.2	1.5	330	65
Block 2	22	1.15	0.25	1	195	55
Block 2	23	0.8	0.2	1.5	330	65
Block 2	24	1.15	0.25	2	465	55
Block 2	25	1.15	0.15	1	465	45
Block 2	26	0.8	0.2	1.5	330	65
Block 3	27	1.5	0.2	1.5	330	45
Block 3	28	0.8	0.2	1.5	330	48
Block 3	29	0.8	0.2	2.5	330	37
Block 3	30	0.8	0.2	0.5	330	37
Block 3	31	0.8	0.2	1.5	60	26
Block 3	32	0.8	0.3	1.5	330	50
Block 3	33	0.8	0.2	1.5	330	53
Block 3	34	0.1	0.2	1.5	330	25
Block 3	35	0.8	0.1	1.5	330	41
Block 3	36	0.8	0.2	1.5	600	12

**Table 4 foods-13-00143-t004:** ANOVA for selected factorial model (Plackett–Burman).

Source	Sum of Squares	df	Mean Square	F-Value	*p*-Value	Source
Model	0.4693	7	0.0670	50.24	<0.0001	significant
A-Sample weight	0.1483	1	0.1483	111.17	<0.0001	
B-Type of solvent	0.0245	1	0.0245	18.38	0.0008	
D-Water volume	0.0081	1	0.0081	6.10	0.0270	
E-AF Weight	0.0093	1	0.0093	6.98	0.0193	
H-Sorbent weight	0.0688	1	0.0688	51.55	<0.0001	
K-Cleanup time	0.0377	1	0.0377	28.28	0.0001	
L-NaCl weight	0.0467	1	0.0467	35.00	<0.0001	
Curvature	0.1682	2	0.0841	63.03	<0.0001	
Residual	0.0187	14	0.0013			
Lack of Fit	0.0111	6	0.0019	1.97	0.1846	not significant
Pure Error	0.0075	8	0.0009			
Cor Total	0.6562	23				

**Table 5 foods-13-00143-t005:** Fit statistics for Plackett–Burman design.

Fit Statistics	Values
R^2^	0.9617
Adjusted R^2^	0.9426
Predicted R^2^	0.8065
Adeq Precision	23.5079

**Table 6 foods-13-00143-t006:** Factors, notation, type, unit, and their levels in central composite design (CCD).

Factor	Notation	Type	Unit	Levels	
				−1 *	+1 **
NaCl weight	A	Numeric	g	0.1	1.50
Sorbent weight	B	Numeric	g	0.1	0.3
Ammonium formate weight	C	Numeric	g	0.5	2.50
Cleanup vortex time	D	Numeric	s	60	600

* −1: Minimum; ** +1: Maximum.

**Table 7 foods-13-00143-t007:** Analysis of variance (ANOVA) results for CCD.

Source	Sum of Squares	df	Mean Square	F-Value	*p*-Value	Source
Block	0.8849	2	0.4424			
Model	0.3526	14	0.0252	139.79	<0.0001	significant
A-NaCl weight	0.0339	1	0.0339	187.91	<0.0001	
B-Sorbent Weight	0.0143	1	0.0143	79.43	<0.0001	
C-AF	5.766 × 10^−7^	1	5.766 × 10^−7^	0.0032	0.9555	
D-cleanup time	0.0274	1	0.0274	151.98	<0.0001	
AB	0.0107	1	0.0107	59.20	<0.0001	
AC	0.0148	1	0.0148	82.08	<0.0001	
AD	0.0063	1	0.0063	34.85	<0.0001	
BC	0.0005	1	0.0005	2.52	0.1288	
BD	0.0047	1	0.0047	26.21	<0.0001	
CD	0.0009	1	0.0009	4.73	0.0425	
A^2^	0.0423	1	0.0423	234.57	<0.0001	
B^2^	0.0042	1	0.0042	23.08	0.0001	
C^2^	0.0341	1	0.0341	189.22	<0.0001	
D^2^	0.1625	1	0.1625	902.26	<0.0001	
Residual	0.0034	19	0.0002			
Lack of Fit	0.0020	10	0.0002	1.29	0.3544	not significant
Pure Error	0.0014	9	0.0002			
Cor Total	1.24	35				

**Table 8 foods-13-00143-t008:** Fit statistics parameters for CCD optimization.

Fit Statistics	Values
R^2^	0.9904
Adjusted R^2^	0.9833
Predicted R^2^	0.9511
Adeq Precision	78.0531

**Table 9 foods-13-00143-t009:** Compounds, retention times, quantification ions, coefficients of determination, LOQ values, linear ranges, recovery%, and %RSD.

Compound	* RT (Min)	** QI	R^2^	LOQ (ng/g)	Linear Range (ng/g)	Recovery% (Mean)	%RSD (Range)
Naphthalene	11.8	128.1	0.999	1	1–40	115.3	10.4–12.8
Acenaphthylene	18.2	152.1	0.999	2	2–40	110.9	7.9–13.1
Acenaphthene	19	153.1	0.999	0.5	0.5–40	95.4	9.5–11.9
Flourene	21.3	166.1	0.999	1	1–40	113.6	11.2–15.3
Phenanthrene	25.9	178.1	0.999	0.5	0.5–40	97.1	5.2–9.8
Anthracene	26.1	178.1	0.999	0.5	0.5–40	119.5	3.2–7.5
PCB 28	27.6	255.9	0.999	0.5	0.5–40	100.4	1.1–2.8
PCB 52	29	291.9	0.999	0.5	0.5–40	101.7	1.5–2
PCB 49	29.2	291.9	0.999	0.5	0.5–40	98.9	1.3–2.1
PCB 44	29.8	291.9	0.999	0.5	0.5–40	100.7	1.8–2.5
PCB 74	31.2	290	0.999	0.5	0.5–40	110.1	1.2–2.1
PCB 66	31.5	291.9	0.999	0.5	0.5–40	103.9	1.2–1.9
Flouranthene	31.7	202.1	0.999	1	1–40	80.9	6.4–10.5
PCB 155	32.1	357.8	0.999	0.5	0.5–40	93.7	1.5–2.4
PCB 101	32.4	325.9	0.999	0.5	0.5–40	110.6	1.1–1.8
PCB 99	32.6	325.9	0.999	0.5	0.5–40	99.4	1.9–2.9
Pyrene	32.7	202.1	0.999	0.5	0.5–40	97.6	1.3–2.5
PCB 112	33.01	325.91	0.999	0.5	0.5–40	119.7	1.1–1.6
PCB 97	33.2	325.9	0.999	0.5	0.5–40	110.8	1.2–1.9
PCB 87	33.4	325.9	0.999	1	1–40	105.8	1.6–2.3
PCB 110	33.8	325.9	0.999	0.5	0.5–40	87.9	1.2–2.1
PCB 151	34.3	359.8	0.999	0.5	0.5–40	93.6	1.6–2.8
PCB 149	34.7	359.8	0.999	0.5	0.5–40	110.9	1.4–2.8
PCB 118	34.8	325.9	0.999	0.5	0.5–40	116.4	1.3–2.6
PCB 146	35.4	359.8	0.999	0.5	0.5–40	105.9	1.9–2.9
PCB 153	35.6	359.8	0.999	0.5	0.5–40	111.8	1.1–2.3
PCB 138	36.7	325.9	0.999	1	1–40	84.2	1.2–1.9
PCB 158	36.8	359.8	0.999	1	1–40	90.5	1.3–2.5
PCB 178	37.02	393.8	0.999	0.5	0.5–40	98.2	1.8–2.7
PCB 187	37.32	393.8	0.999	0.5	0.5–40	94.3	1.3–2.3
PCB 183	37.5	393.8	0.999	0.5	0.5–40	101.1	1.2–1.8
PCB 128	37.8	359.8	0.999	0.5	0.5–40	93.8	1.5–2.1
PCB 177	38.4	393.8	0.999	0.5	0.5–40	105.7	1.3–2.1
Benzo[a]anthracene	38.5	228.1	0.999	0.5	0.5–40	110.6	8.8-12.9
Chrysene	38.7	228.1	0.999	0.5	0.5–40	113.1	8.3–14
PCB 172	38.9	393.8	0.999	0.5	0.5–40	116.4	1.2–2.9
PCB 180	39.17	393.8	0.999	0.5	0.5–40	93.7	1.7–2.7
PCB 170	40.2	393.8	0.999	0.5	0.5–40	109	1.9–2.5
PCB 198	40.4	429.8	0.999	0.5	0.5–40	99.5	1.5–2.3
PCB 199	40.5	429.8	0.999	0.5	0.5–40	89.9	1.2–2.5
PCB 203	40.7	429.8	0.999	0.5	0.5–40	93.8	1.8–2.7
PCB 196	40.9	429.8	0.999	0.5	0.5–40	111	1.9–2.8
PCB 195	41.9	429.8	0.999	0.5	0.5–40	103	1.6–2.7
PCB 194	42.65	429.8	0.999	0.5	0.5–40	96.4	1.3–2.7
Benzo[b]flouranthene	43.9	252.1	0.999	0.5	0.5–40	80.6	8.9–12.2
Benzo[k]flouranthene	44.09	252.1	0.999	0.5	0.5–40	75.4	10–14.2
PCB 206	44.3	463.8	0.999	0.5	0.5–40	81.5	1.6–2.8
PCB 209	45.9	497.7	0.999	0.5	0.5–40	81	1.8–2.9
Benzo[a]pyrene	45.9	252.1	0.999	0.5	0.5–40	79	10.2–14.1
Indeno[1,2,3-cd]pyrene	54.9	276.1	0.999	1	1–40	75	12.6–16.1
Dibenz[a,h]anthracene	55.17	278.1	0.999	1	1–40	75.6	11.1–15.7
Benzo[ghi]perylene	57.5	276.1	0.999	2	2–40	72.5	12.6–16.8

* Retention Time; ** Quantifier Ion.

**Table 10 foods-13-00143-t010:** Comparison of the LOQ, recovery%, %RSD, and extraction method for simultaneous analysis of PAHs and PCBs in meat matrices by GC-MS.

Sample	Analyte	LOQ (ng/g)	Recovery% (Range)	%RSD (Range)	Extraction Method	References
Smoked meat products and liquid smokes	fluoranthene, pyrene, benzo[a]anthracene, chrysene, benzo[b]fluoranthene, benzo[k]fluoranthene, benzo[a]pyrene, indeno [1,2,3-c,d] pyrene, dibenzo [a,h]anthracene and benzo[g,h,i]perylene)	0.01	90–110	3–12	ASE	[39]
High-fat salmon	(PAHs)	-	50–200	<10	Modified QuEChERS extraction, dispersive SPE	[56]
Processed meats: sausages, cerveroni pepperoni, beer pepperoni, hams, mortadellas, smoked ribs, Antioquia’s sausages, bacon	PAHs	0.5	55–116	≤20	QuEChERS method	[57]
Smoked meat products	16 common PAHs	1 and 10	74–117	1.15–37.57	Development of QuEChERS method	[58]
Smoked meat	PAHs	0.5	63–94%	-	SFE-plus-C18	[59]
The smoke of charcoal-grilled meat restaurants	PAHs	0.075–15	80–99 for the 50 ppb; 81–100 for the 250 ppb; 82–101 for the 750 ppb	-	QuEChERS method	[60]
Chinese mitten crabs	PCBs	0.1–3.6	85.9–119.8	-	QuEChERS	[61]
Daily consumed fish (Grass Carp)	PAHs–PCBs	0.6	98–95	1.02–8.9	Microwave-assisted extraction (MAE), lip removal by gel permeation chromatography (GPC), cleanup by solid-phase cartridge (SC)	[62]
Grilled meat sausages	16 (PAHs)	0.20	60–134 at the 0.5 µg/kg and 69–121 at the 1.0		saponification, liquid–liquid partition, and final cleanup using solid-phase extraction (SPE)	[63]
Dry pork neck (budiola), pork tenderloin, and sausages	PCBs	5.7–13.8	81.61–116.33	3.7–10	QuEChERS	[64]
Grilled meat	16 PAHs, 36 PCBs	0.5–2 0.5–1	72.5–119.5 81–116.4	1.3–16.8 1.2–2.9	Modified QuEChERS	Proposed study

## Data Availability

Data is contained within the article.

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
