# Peer review of "Multivariate Optimization and Validation of a Modified QuEChERS Method for Determination of PAHs and PCBs in Grilled Meat by GC-MS"

_foods, 2023, doi:10.3390/foods13010143_

Round 1

Reviewer 1 Report

The authors present a clear abstract stating their work's objective, highlighting essential results, and discussing the scope of their research for future investigations. However, they should include more precise conclusions that emphasize the application of their developed method and the importance of monitoring polycyclic aromatic hydrocarbons (PAHs) and polychlorinated biphenyls (PCBs) in human health and the environment. PAHs and PCBs are persistent and toxic chemical compounds in various products and human activities. Their presence in the environment poses risks to health, including carcinogenic and endocrine-disrupting effects. Therefore, emphasizing the significance of monitoring and developing effective methods for detecting and quantifying these contaminants is crucial for protecting public health and the ecosystem. By highlighting the applicability of their method and its contribution to the detection and analysis of PAHs and PCBs, the authors will strengthen the relevance and impact of their research.

The introduction provided by the authors is good and covers the most critical points regarding contamination with PAHs and PCBs. They also emphasize the high toxicity of these compounds. The authors discuss the advantages and disadvantages of analytical techniques and propose using QuEChERS as an analytical alternative. They properly justify the application of QuEChERS in the context of environmental contamination and environmental chemistry. All of the above should be retained in the introduction. However, the authors must mention why they specifically analyze grilled meat or food prepared on a grill. As a result, readers are left speculating that cooking food using this method may lead to contamination, but this needs to be made clear. Therefore, the authors should allocate one or two paragraphs in their introduction to clarify this situation and provide the current state of the literature on the topic.

Figure 1 needs to improve the resolution.

Figures 2 and 3 present precise information and are understandable, but their resolution is low, resulting in pixelation. Please improve this aspect by enhancing the resolution of the figures.

The authors should review the journal's template again, as I noticed that the tables have different font sizes, and the conclusions have different typography and size. Please take the time to ensure consistency in the font type and size throughout your manuscript.

The conclusions, like the introduction, are excellent and clear, addressing the research question and objective posed by the authors. However, they are solely focused on the analytical aspect of the study. The authors should keep in mind that the journal covers this aspect within its scope, but it should also consider the impact on food and its relationship with human health. Please include these aspects in both sections to provide a more comprehensive and well-rounded conclusion.

The work is interesting, aligns with the scope of the journal, provides solid results, and has an excellent discussion. The manuscript should be considered  if the authors address the previous comments. Therefore, I suggest major revisions.

Author Response

Thank you for your comment. 

The revised version of manuscript according to your useful comments is prepared and the manuscript in new versions is ready  to review.

Reviewer 2 Report

The article "Multivariate optimization and validation of a modified QuEChERS method for determination of PAHs and PCBs in grilled meat by GC-MS" presents a study focused on optimizing and validating a modified QuEChERS method for the analysis of polycyclic aromatic hydrocarbons (PAHs) and polychlorinated biphenyls (PCBs) in grilled meat using gas chromatography-mass spectrometry (GC-MS). The authors aim to address the persistent organic pollutant (POP) concerns associated with PAHs and PCBs in food. However, there are several areas that require improvement in terms of clarity, organization and depth of analysis.

Novelty and Aim: The authors should clearly state the novelty of their work, highlighting how it contributes to the existing knowledge on the subject matter.

Abstract: In the abstract, the authors state that the method "could minimize matrix interference". Is it an assumption or a result based on the study outcomes?

Introduction: The introduction section needs reorganization. Consider moving the paragraph from lines 58-65 before the paragraph from lines 41-57. Additionally, the authors should provide a comprehensive analysis of the regulatory levels beyond the EC maximum limits (MLs) to offer a broader perspective.

The term "QuEChERS" is mentioned for the first time in the introduction (line 72), so the authors should provide a brief explanation of the abbreviation. The same applies to the abbreviation "LOQ" (limit of quantification) in line 25.

Methods and Materials: The absence of a "Methods and Materials" section is a significant drawback, as it is essential for providing details on how the study was conducted. The absence of the section makes it challenging to understand critical aspects of the study, such as sampling details, the materials used, and the specific methodology employed. Including this section is essential to ensure transparency and reproducibility. It is crucial to include this section in scientific works.

Reorganization and References: The article lacks proper usage of up-to-date references and fails to adequately compare the results obtained in this study with relevant findings from other similar scientific works. Citing appropriate references throughout the article, especially when presenting information not obtained by the authors, is crucial to enhance the credibility of the study. For example, the paragraph from lines 192-202 should have a reference unless the information presented is derived from the author's own findings.

Conclusion: The main advantages of the validated method should be emphasized, while the study limitations should be acknowledged in the conclusion section. Currently, the conclusion is weak and requires improvement to effectively summarize the findings and their implications.

Overall, the article shows promise in addressing the optimization and validation of a modified QuEChERS method for the analysis of PAHs and PCBs in grilled meat. However, to enhance its quality, the authors need to address the aforementioned areas of improvement, including the inclusion of a "Methods and Materials" section, clarity on the novelty of the work, proper organization, accurate references, and a more robust conclusion section. By addressing these points, the article can contribute significantly to the field of food contaminant analysis and detection.

The overall quality of the English language in the provided text is good. However, there are a few areas where improvement can be made to enhance clarity and conciseness.

Author Response

(The authors gave the same response as above.)

Reviewer 3 Report

In the paper Multivariate optimization and validation of a modified QuEChERS method for determination of PAHs and PCBs in grilled meat by GC-MS, authors report a validation method for polluttants quantification. 

Paper is interesting hpwever several part must be emproved.

1. Introduction 

Persistent organic pollutants (POPs) as compounds of anthropogenic,have high lipophilicity direction to the bioaccumulation in living organisms' fatty tissues. 

Please modify, consider that PAH can by orifinated naturally. In this context add several references such as 

A Short Review of Simple Analytical Methods for the Evaluation of PAHs and PAEs as Indoor Pollutants in House Dust Samples. Atmosphere, 13(11), 1799.

Two groups of these compounds are polycyclic aromatic hydrocarbons (PAHs) and polychlorinated biphenyls (PCBs), and their 

persistence in the environment is a critical issue due to their toxic effec and persistence [1-2].

To underline the aspect of persistence, please add some relevant references that report stability of PCB.

Multi contaminants methods are used in foods, especially for determining PAHs and PCBs, primarily according to gas chromatography-mass spectrometry. (GC-MS) [5-7], GC-flame ionization detector (GC-FID) [8], GC-electron capture detector (GC-ECD) [1-9], gas chromatography-tandem mass spectrometry 

(GC-MS/MS) [1,10,11], liquid chromatography-tandem mass spectrometry (LC-MS/MS) [12-13], gas chromatography-high resolution mass spectrometry [14-15], and supercritical fluid chromatography/atmospheric pressure chemical ionization-mass spectrometry (SFC/APCI-MS) [15]. 

This sentence is not clear....

English must be emproved

Author Response

(The authors gave the same response as above.)

Round 2

Reviewer 3 Report

Changes were mades.  

Only few part can be emproved before publication. 

From line 45 to line 51

Pleas add reference such as 

Profiles and sources of PAHs in sediments from an openpit mining area in the Peruvian andes.

To underline the photostability of PCB add reference concerning the photochemical sample treatment for PCB analysis. 

English is fine